# Chiral BINOL-phosphate assembled single hexagonal nanotube in aqueous solution for confined rearrangement acceleration

Kang Li [1,2,3] ✉, Wei-Min Qin[1], Wen-Xia Su[1], Jia-Min Hu[1] & Yue-Peng Cai [1,2,3] ✉

Creating microenvironments that mimic an enzyme's active site is a critical aspect of supramolecular confined catalysis. In this study, we employ the commonly used chiral 1,1'-bi-2-naphthol (BINOL) phosphates as subcomponents to construct supramolecular hollow nanotube in an aqueous medium through non-covalent intermolecular recognition and arrangement. The hexagonal nanotubular structure is characterized by various techniques, including X-ray, NMR, ESI-MS, AFM, and TEM, and is confirmed to exist in a homogeneous aqueous solution stably. The nanotube's length in solution depends on the concentration of chiral BINOL-phosphate as a monomer. Additionally, the assembled nanotube can accelerate the rate of the 3-aza-Cope rearrangement reaction by up to 85-fold due to the interior confinement effect. Based on the detailed kinetic and thermodynamic analyses, we propose that the chain-like substrates are constrained and pre-organized into a reactive chair-like conformation, which stabilizes the transition state of the reaction in the confined nanospace of the nanotube. Notably, due to the restricted conformer with less degrees of freedom, the entropic barrier is significantly reduced compared to the enthalpic barrier, resulting in a more pronounced acceleration effect.

Supramolecular confined catalysis involves the use of carefully designed microenvironments that mimic the active pocket of an enzyme to enhance the efficiency and selectivity of catalytic reactions. This strategy enables precise control of reaction parameters, such as substrate concentration, orientation, and access to catalytic sites, resulting in more effective catalysis[1–4]. As a result, the construction of advanced supramolecular catalysts with intricate confined spaces, similar to artificial enzymes, is a critical issue in this area. To achieve this goal, a variety of synthetic systems, such as organic hosts (e.g., macrocycles[5–8], tubes[9–13], capsules[14–19]) and coordination cages in homogeneous solutions[20–28], as well as metal-organic frameworks (MOFs) in heterogeneous phases[29,30], have been designed and furnished.

The field of organocatalysis has achieved significant progress over several decades, with certain organocatalysts demonstrating catalytic efficiency that matches or even surpasses that of enzymes[31–33]. Notably, the concept of confined catalysis has been discussed in the context of certain organocatalysts, wherein the active site is buried within a well-defined space through elegant catalyst design[34,35]. However, developing more complex and organized structures to mimic enzyme function through covalent bonding approaches remains a challenging and intricate endeavor. Alternatively, supramolecular self-assembly offers a simple and efficient means of creating advanced platforms through noncovalent interactions. Incorporating organocatalysts as subcomponents into these assemblies presents an attractive approach. By combining the catalytic properties of organocatalysts with precisely designed nanospaces, complementary and synergistic effects could be generated. Although organocatalysts have been integrated into MOF materials for heterogeneous catalysis[36–39], their assembly in homogeneous solution conditions is rare and requires further exploration.

[1]School of Chemistry, South China Normal University, Guangzhou 510006, China. [2]Guangzhou Key Laboratory of Energy Conversion and Energy Storage Materials, Guangzhou 510006, China. [3]The Joint Laboratory of Energy Materials Chemistry for SCNU and TINCI, Guangzhou 510006, China. ✉e-mail: likang5@m.scnu.edu.cn; caiyp@scnu.edu.cn

The chiral BINOL-phosphate derived Brønsted acid is a kind of versatile organocatalyst widely used in asymmetric catalysis[40–42]. Commonly, aryl panels are used as pending branches to be installed at the ortho-position of the phosphoric acid group, resulting in a molecular clip structure with an unfolded angle of approximately 120° (Fig. 1a)[43]. This defines a semi-closed space with a wide open on its mirror side. Based on the geometrical analysis of the BINOL-phosphate motif, rhombus (dimer) and hexagon (hexamer) structures can be constructed in a plane to create more confined space (Fig. 1b). Actually, synthetic chemists have successfully obtained the dimer form through covalent bonding. For instance, List's group employed the imidodiphosphate moiety to bond two identical BINOL subunits together[34], while Gong's group used the alkyl ether as a spacer to bridge them[44]. These dimer-like organocatalysts have exhibited highly catalytic activity and selectivity, with enzyme-like properties arising from the well-defined and sufficient catalytic space. Otherwise, for the more complicated hexagon model, the covalent bonding strategy appears to be insufficient. Alternatively, we used the self-assembly strategy to build the desired structure in a supramolecular manner. In this study, we investigated the self-assembly behavior of the enantiomers of BINOL-derived phosphoric acid (referred to as R/S 4-hydroxy-2,6-di-2-naphthalenyl-4-oxide-dinaphtho[2,1-d:1',2'-f][1,3,2]dioxaphosph-epin, denoted as R/S-HNOP) in aqueous solution.

## Results

### Self-assembly and characterization of nanotube

In practice, a DMSO solution of R/S-HNOP was treated with 1 equiv. of NaOH to neutralize the phosphoric acid group, followed by the addition of water to drive the self-assembly process through intermolecular arrangement (Fig. 2a). Fortunately, the single crystals of both enantiomers suitable for X-ray analysis were obtained from their corresponding aqueous solution (15 mM) after 5 days (Supplementary Tables 1–3, Supplementary Fig. 6). Taking R-HNOP as an example, it crystallizes in the $P6_5$ space group. Six R-HNOP molecules form a hexagon by spanning the edges (Fig. 2b). In addition to π-π stacking and overlap of intermolecular 2-naphthyl groups, the geometric constraint of molecular configuration contributes to the spatial arrangement during assembly. Crystal structure analysis showed that the hexagon-shaped ensemble has an inner confined cavity measured to be 1.5 nm x 1.4 nm, with great potential for host-guest interactions. As a repeat unit, the hexagon-shaped ensemble stacks closely to form a nanotube along the c-axis, which has a uniform channel across its

interior and an exterior size of approximately 3.0 nm (Fig. 2c). Two 6-fold screw axes ($6_5$, $6_1$) are located in the central nanotube along the c-axis, generating repeated hexagons and five strands of right-handed helix chain (Fig. 2d).

More importantly, we aimed to investigate the assembly behaviors of deprotonated R/S HNOP (NOP⁻) and verify the existence of assembled hexagon or nanotube in homogeneous aqueous environment. The nuclear magnetic resonance (NMR) spectra of R-NOP⁻ were examined in both DMSO-$d_6$ and DMSO-$d_6$/$D_2O$ (1/2) mixture (Supplementary Figs. 1–4). The ¹H NMR spectrum showed sharp and well splitted peaks in the 7.2 ppm–8.4 ppm region in DMSO-$d_6$, indicating a well-dissolving and molecular-level dispersion state. However, when dissolved in DMSO-$d_6$/$D_2O$ (1/2) mixture, the chemical shifts of all protons moved to upfield, with an average shift of ca. 0.3 ppm, except for peak g and h/h' which gave relatively large upfield shifts to 0.61 ppm and 0.68 ppm, respectively (Fig. 3a). This uniform upfield shift could be explained by the close packing of R-NOP⁻ in aqueous solution owing to the geometry- directed molecular recognition. The weak shielding effect resulted in a small upfield shift of protons, while the relatively large shift (near to 0.7 ppm) of peak g and h/h' indicated a stronger shielding effect due to intermolecular close packing. This is consistent with the crystal structure, where the naphthalene ring is deeply buried in the interval of adjacent aromatic panels with more closer packing mode, while the binaphthyl backbone is located in the exterior of the hexagon. Based on the NMR analysis, it can be inferred that the R-NOP⁻ assembled hexagon or subcomponent of nanotube may exist in aqueous solution.

In the negative electrospray ionization time-of-flight mass spectrometry (ESI-TOF-MS) spectrum (Fig. 3b, Supplementary Fig. 5), species ranging from dimer to pentamer were identified, suggesting an oligomeric existing state for R-NOP⁻ in aqueous solution under MS conditions. The isotopic pattern of the pentamer closely matched the simulated pattern (m/z, found: 3089.6684, simulated: 3089.6719). Although even higher oligomers such as hexamer were not detected, it is likely that the noncovalent assembled hexagon or nanotube is fragile and prone to dissociation under MS conditions. Nevertheless, these fragmented oligomers provide valuable evidence to disclose the assembly process of nanotube in aqueous solution.

To further detect the assembled form of R-NOP⁻ in solution, we conducted an in situ atomic force microscopy (AFM) test in wet conditions (Fig. 3c, Supplementary Fig. 8). A mixture solution of R-NOP⁻ (15 mM, DMSO/$H_2O$ = 1/2) was deposited onto a mica plate, and distinct single nanotubes were detected on the AFM images[18]. The height of the nanotubes was measured using two section lines (lines a and b) across different areas. Line a showed a uniform height of approximately 3.0 nm, which was consistent with the crystal structure results (the radial direction size of the hexagonal nanotube measured to be 3.0 nm). Line b passed through one separated and two intersected nanotubes, giving a height of approximately 3.0 nm for the former nanotube and approximately 6.0 nm for the overlaid pair, indicative of the uniformity of the single nanotube. With this convincing evidence, we could conclude that the R-NOP⁻ underwent a self-assembly process to form a stable single hexagonal nanotube in aqueous solution. In addition, when a red laser was shone through R-NOP⁻ aqueous solutions, a bright light path was observed due to the Tyndall effect (Fig. 3d), ascribed to the presence of assembled nanotubes in the specific aqueous environment[45]. Meanwhile, the transmission electron microscope (TEM) methodology revealed a fiber-like morphology with an average width of approximately 10 nm (Supplementary Fig. 13), which was likely grown from the arrangement and packing of single nanotube under dry conditions.

We also explored the average distribution of nanotube length against concentration parameter by diffusion-ordered spectroscopy (DOSY) measurement. With a series of obtained diffusion coefficient (D) values under different NOP⁻ concentration conditions as input

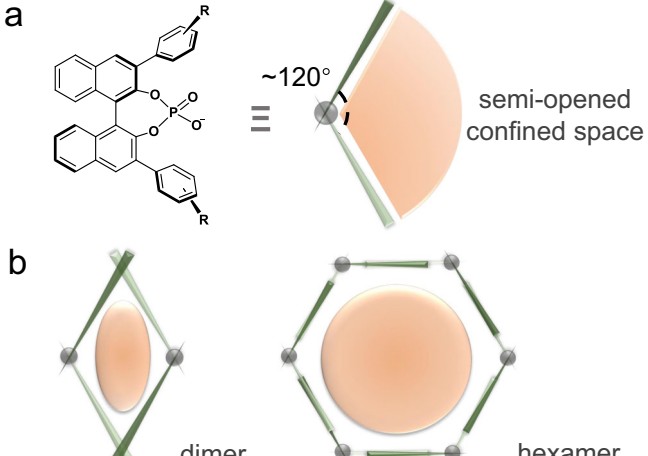

**Fig. 1 | Comparison of confined space by BINOL-organocatalyst. a** $C_2$ symmetric BINOL-phosphate with a semi-opened confined space. **b** Geometry-directed assembly models based on BINOL-phosphate containing well-defined compact space. (The light-orange shapes represent the confined space).

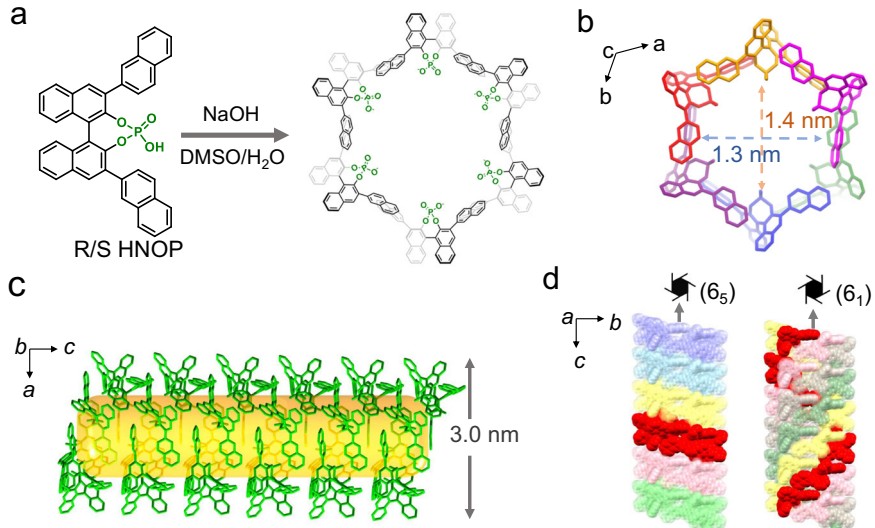

**Fig. 2 | Representative of the self-assembly process and crystal structure analysis. a** Self-assembly of the hexagonal ring with chiral BINOL-phosphate acid under the basic condition in aqueous solution. **b** The crystal structure of the hexagon viewed along the *c* axis with labelled inner size (R-nanotube). **c** The hexagonal ring packing to the nanotube viewed along the *b* axis (the yellow color represents the interior channel). **d** The space-filling nanotube holding two screw axes across the inner center viewed along the *a* axis (the red color highlights a hexagon or single helix chain corresponding to the screw axis).

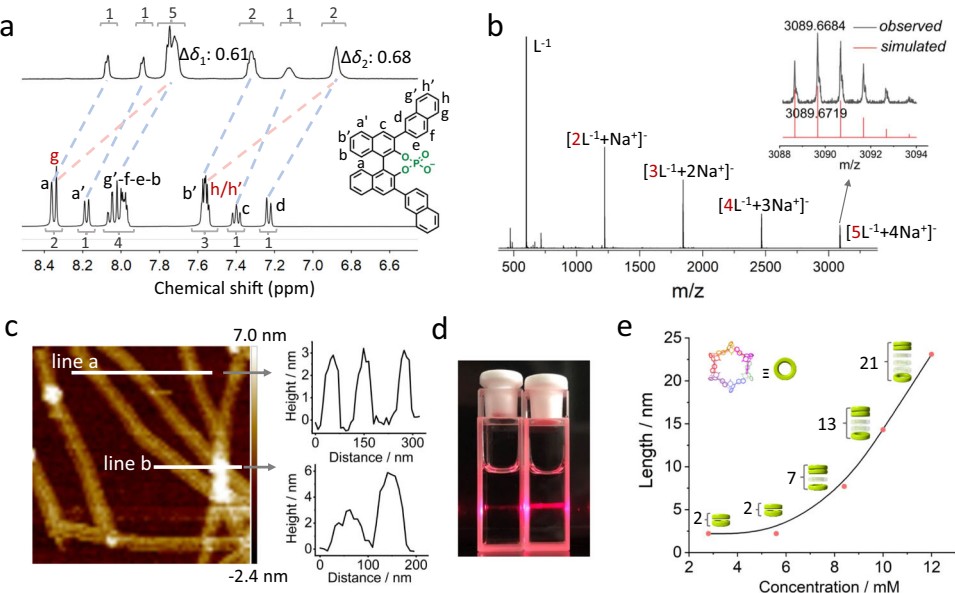

**Fig. 3 | Characterization of nanotube in solution. a** Comparison of partial ¹H NMR spectra (400 MHz, 298 K) of NOP⁻ in DMSO-$d_6$ and DMSO-$d_6$/D₂O mixture (v:v = 1:2). **b** The negative mode ESI-TOF-MS spectrum of NOP⁻ in aqueous condition, inserted is the enlarged assignment of pentamer. **c** In situ AFM image of NOP⁻ in 1:2 DMSO/H₂O mixture, two distinct regions were selected along the white lines and their corresponding heights were measured and plotted on the right. **d** Photograph of a couple of NOP⁻ solutions in DMSO (left) and 1:2 DMSO/H₂O mixture (right) under irradiation of a beam of red laser. **e** Monomer (NOP⁻) concentration-dependent length of the single nanotube with different layers of assembled hexagon in solution (the green torus represents assembled hexagon using NOP⁻ monomer as one layer; the figure beside each stacked torus represents the layer number; the last three stacked torus were not drawn as their real layer number and simplified for clarity.).

(Supplementary Fig. 9), we could calculate the corresponding stacking layer number of hexagon unit and nanotube length by modified Stokes-Einstein equation (Supplementary Eq. 1) with cylinder model and bead model, respectively (Supplementary Figs. 10–12, Supplementary Tables 4, 5)[46,47]. As a result, the size of the nanotube is closely related to the concentration of NOP⁻ aqueous solution (Fig. 3e). At a relatively low concentration region (2.8–5.6 mM), a short nanotube consisting of only two layers of assembled hexagon is formed (*ca.* 2.2 nm of length). As the gradual increase of concentration from 5.6 mM to 12.0 mM, the length of the nanotube grows up to 23 nm consisting of *ca.* 21 layers of the hexagon. At even higher concentration conditions, the solution becomes viscous and loses a certain extent of fluidity, indicating the existence of a longer nanotube.

## Catalytic investigation of assembled nanotube in solution

Considering the BINOL-phosphate molecular scaffold as a privileged type of organocatalyst and the creation of a well-defined nanospace in our case, the supramolecular nanotube was used for catalytic investigation. We chose the 3-aza-Cope rearrangement[48] as the model reaction for two reasons: firstly, the quaternary ammonium salt as

substrate is highly soluble in aqueous solution, thus facilitating homogeneous catalysis; secondly, the nanotube's interior bears highly negative charge (phosphate group) and can form tight ion-pairing with cationic substrates, allowing for effective confined catalysis. The reaction in aqueous medium occurs in two steps: the [3,3] sigmatropic rearrangement takes place to form an iminium as cation intermediate, which has a relatively high Gibbs energy barrier; then, the unstable intermediate is rapidly hydrolyzed to the corresponding unsaturated aldehyde. As a result, the first procedure is the rate-determining step, which controls the overall reaction rate.

We synthesized two chain-like quaternary ammonium bromides as substrates that contain olefinic bonds on both sides. The derived substrate 2 (S2) with methyl-substitution is bulkier and has more steric hindrance than substrate 1 (S1). To facilitate the efficient host-guest recognition and exchange between nanotube and substrates, the short nanotube with dual layers was applied for the following host-guest interactions and catalytic study. The isothermal titration calorimetry (ITC) experimental result gave the binding constants of $1.08 \times 10^4\,M^{-1}$ for S1 and $9.41 \times 10^4\,M^{-1}$ for S2, respectively, indicating that the bulkier S2 forms a more tight ion-pair with nanotube than S1 (Supplementary Fig. 14). Additionally, the short nanotube could approximately bind three guest molecules per hexagon sub-component for both substrates. The chemical shifts of substrates coincidently moving upfield (an average of *ca.* 0.2 ppm) under nanotube conditions also supported the host-guest interaction behavior.

As shown in Table 1, initially, we tested the background reaction rates of both substrates (No. 1, 6). The observed kinetic data is fitted well with the first-order reaction rate law (Supplementary Figs. 15, 17, 21, 23). However, both substrates have slow reaction rates even at elevated temperatures, which suggests the need for an effective catalyst to accelerate the reaction rates. When the assembled nanotube was introduced as a homogeneous catalyst (Cat1 nanotube), the reaction rates of both substrates were enhanced (No. 2, 7 and Supplementary Figs. 16, 18, 19, 22, 24, 25). S2 exhibited a more significant rate enhancement effect ( ~ 85 fold) than S1 ( ~ 17 fold), probably due to

better steric control for intramolecular intertangling in a confined nanospace. We will discuss this in detail later.

To elucidate the origin of the acceleration effect by the Cat1 nanotube, we conducted a series of control experiments (Supplementary Figs. 20, 26). Initially, DMSO-$d_6$ was used as a solvent instead of DMSO-$d_6$/D$_2$O mixture (No. 3, 8). In this case, R-NOP$^-$ (Cat1) would be molecular solvated without assembled nanotube in solution. Cat1 in DMSO exhibited some extent of acceleration for both substrates, albeit not as notably as Cat1 nanotube in aqueous solution. This discrepancy could be attributed to the semi-opened space created by Cat1 itself may not effectively constrain and pre-organize the substrates near the active site, resulting in a suboptimal level of activation toward the transition state of reactions. Secondly, we employed the crystalline state of Cat1 to catalyze the corresponding reactions in a heterogeneous fashion (No. 4, 9 and Supplementary Fig. 7). The results showed a negligible acceleration for both substrates compared to the background reactions. This could be explained that the close packing of the nanotube in the crystalline state with millimeter size reduces the efficiency of the mass transfer and exchange process under macroscale. In contrast, the monodispersed Cat1 nanotube in a homogeneous solution enables highly efficient substance diffusion and exchange, facilitating the capture of substrates and release of products to fulfill catalytic turnover. Thirdly, an alternative catalyst, Cat2 was employed (No. 5, 10), which has no substituted group on the naphthalene panel to create a potential confined space. NMR analysis of Cat2 in aqueous solution also revealed no self-assembly behavior. As expected, the acceleration effect for S1 and S2 under Cat2 conditions was limited. Hence, we could conclude that the significant acceleration effect of Cat1 nanotube is attributed to its excellent homogeneous monodispersion and the well-defined nanospace created to modulate the reactivity of substrates.

By utilizing the Eyring equation (Supplementary Eq. 2) derived from transition state theory, it is possible to correlate the kinetic reaction rate with the typical thermodynamic parameters[49]. With this analytical method, we obtained two sets of thermodynamic data for both substrates under free and Cat1 nanotube catalytic conditions

**Table 1 | Acceleration effect for 3-aza-Cope rearrangement by catalysts under different conditions[a]**

R: H (S1)
R: Me (S2)

intermediate

Cat1:

Cat2:

| No. | R | Catalyst | Solvent | $k_{obs}$ $10^{-5}\,s^{-1}$ | Acceleration |
|-----|-----|----------|---------|------------------------------|--------------|
| 1 | H | Blank | DMSO-$d_6$/D$_2$O[b] | 2.6 | —— |
| 2 | H | Nanotube | DMSO-$d_6$/D$_2$O | 43.5 | 17 |
| 3 | H | Cat1 | DMSO-$d_6$ | 6.0 | 2.3 |
| 4 | H | Crystal[c] | DMSO-$d_6$/D$_2$O | 2.8 | 1.1 |
| 5 | H | Cat2 | DMSO-$d_6$/D$_2$O | 4.2 | 1.6 |
| 6 | Me | Blank | DMSO-$d_6$/D$_2$O | 6.4 | —— |
| 7 | Me | Nanotube | DMSO-$d_6$/D$_2$O | 545.0 | 85 |
| 8 | Me | Cat1 | DMSO-$d_6$ | 32.6 | 5.1 |
| 9 | Me | Crystal | DMSO-$d_6$/D$_2$O | 8.1 | 1.3 |
| 10 | Me | Cat2 | DMSO-$d_6$/D$_2$O | 15.3 | 2.4 |

[a]Reaction conditions: substrate (2.8 mM), catalyst (5.6 mM), solvent (600 μL), pD = 8.0, 50 °C, the reaction was conducted in NMR tube and monitored by NMR spectroscopy.
[b]DMSO-$d_6$/D$_2$O: v/v = 1/2.
[c]The nanotube crystal cannot redissolve into the aqueous solution.

(Table 2 and Supplementary Figs. 27, 28). The positive $\Delta H^{\ddagger}$ values indicated the endothermic character of the reactions, meaning that an elevated temperature would accelerate the intramolecular rearrangement. Meanwhile, the negative $\Delta S^{\ddagger}$ values implied that the reaction proceeds through a highly organized transition state with a low degree of freedom. In this reaction, a transient chair-like conformation is proposed to happen for the starting chain-like substrates. In comparison to uncatalyzed reactions, the employment of a nanotube catalyst resulted in a marginal impact on the activation energy of $\Delta H^{\ddagger}$ with no reduction greater than 1 e.u. observed for either substrate. In contrast, the activation energy of $\Delta S^{\ddagger}$ is significantly altered under nanotube catalytic conditions. The entropic barrier of **S1** and **S2** is reduced by 4.3 e.u. and 6.2 e.u., respectively. At this stage, we could conclude that the significant change of $\Delta S^{\ddagger}$ is the key factor resulting in rate acceleration by the Cat1 nanotube. This is mainly due to the effective confinement effect provided by the well-defined nanospace of the nanotube interior. Once the positive substrates are captured by the negative cavity of the nanotube, the outstretched chain-like molecules are forced to intertangle themselves to form chair-like conformers in the confined nanospace, thereby stabilizing the transition state of the intramolecular rearrangement. This was confirmed by the nuclear overhauser effect spectroscopy (NOESY) measurement. Especially for **S2**, the terminal methyl protons have a considerable correlation with the methyl and alkenyl protons at the other end in the NOESY spectrum

under host-guest conditions, which means that the terminal protons with long-distance are pre-organized and confined to be in close proximity in the interior (Supplementary Fig. 29).

Notably, the chair-like conformer with high tension is disfavored and hard to occur under free conditions. However, this highly reactive transition state can be induced and stabilized in the confined microenvironment, thereby lowering the activation barrier of entropy. Specifically, due to the relatively larger size and more steric hindrance of **S2** than **S1**, it forms a more tight ion-pair with less degree of freedom within the cavity for **S2** as revealed by the former ITC and NOESY results, accounting for a more substantial entropy barrier decrease and corresponding more significant rate acceleration effect.

Based on the kinetic and thermodynamic analyses presented above, thus, we propose a catalytic cycle consisting of three main steps (Fig. 4): firstly, the hollow nanotube, with its highly negatively charged confined cavity, captures the cationic substrates by electrostatic interactions[50] and pre-organizes them into a disfavored chair-like conformation; secondly, the constrained substrates undergo a [3,3] sigmatropic rearrangement, resulting in the formation of iminium intermediates with an accelerated reaction rate; thirdly, the unstable intermediates are rapidly hydrolyzed to their corresponding aldehydes in aqueous solution, which allows additional substrates to enter the cavity prior to neutral products, thus enabling the next catalytic cycle.

Besides the rate acceleration effect, we preliminary explored the asymmetric catalytic performance of the chiral nanotube. A substrate with ethyl-substitution on the *trans*-position of terminal olefin was synthesized, which can generate a potential chiral carbon center after catalytic transformation (Supplementary Figs. 30, 31). Finally, it gave a moderate enantioselectivity (around 43% enantiomeric excess) under R-nanotube or S-nanotube catalytic conditions (Supplementary Fig. 32), which showed great potential and improved space by this chiral nanotubular structure for the next asymmetric catalytic investigation.

## Discussion

We have successfully adopted self-assembly and intermolecular interactions to construct a supramolecular hexagonal nanotube based on chiral BINOL-phosphate organocatalyst as subcomponent.

**Table 2 | Thermodynamic parameters calculated by Eyring equation**[a]

| Substrate | Catalyst | $\Delta H^{\ddagger}$ (kcal mol$^{-1}$) | $\Delta S^{\ddagger}$ (cal mol$^{-1}$ K$^{-1}$) | $\Delta\Delta H^{\ddagger}$ (e.u.)[b] | $\Delta\Delta S^{\ddagger}$ (e.u.) |
|---|---|---|---|---|---|
| S1 | Blank | 21.4 (0.2) | −13.2 (0.7) | —— | —— |
|  | Nanotube | 21.0 (0.6) | −8.9 (1.9) | −0.4 | 4.3 |
| S2 | Blank | 21.2 (0.3) | −12.3 (1.1) | —— | —— |
|  | Nanotube | 20.3 (0.4) | −6.1 (1.4) | −0.9 | 6.2 |

[a]Eyring equation: $\ln\frac{k}{T} = \frac{-\Delta H^{\ddagger}}{R}\frac{1}{T} + \ln\frac{k_B}{h} + \frac{\Delta S^{\ddagger}}{R}$ (2). [b]e.u. Equivalent unit.

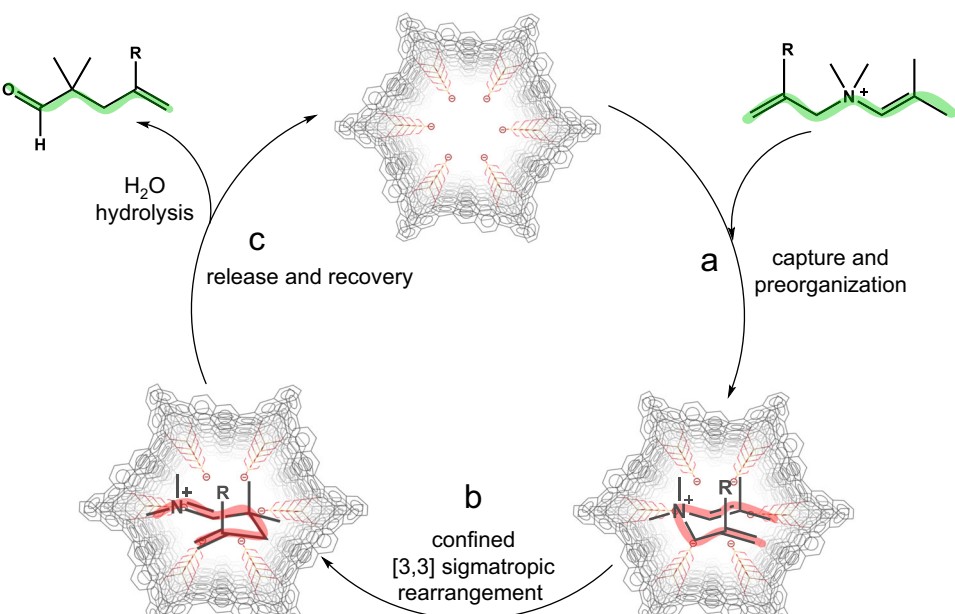

**Fig. 4 | Proposed catalytic cycle of intramolecular rearrangement by homogeneous supramolecular nanotube. a** Cationic substrate captured and pre-organized by the nanotube's cavity (the green wavy line along with the substrate represents its chain-like conformation in free state.). **b** The confined substrate underwent intramolecular rearrangement (the red folding curve along with the molecular structures represents their bent conformation.). **c** The rearrangement product further hydrolyzed to departure from the cavity that making the recovery of the catalytic cycle.

The nanotube stably existed in aqueous solution with a uniform cavity and its axial length is dependent on the concentration of monomer. It was found that the assembled nanotube could act as a homogeneous catalyst to accelerate the 3-aza-Cope rearrangement, with up to an 85-fold acceleration effect in reaction rate. Our analysis showed that the activation energy, enthalpy, and entropy barriers are reduced under nanotube conditions by host-guest interactions, particularly for entropic parameter. The significant reduction in entropic barrier is due to the substrates being forced to adopt a chair-like conformation and losing a certain degree of freedom. These findings provide a new mind for the organocatalyst design principle, not only depending on covalent bond connection but also on the supramolecular self-assembly approach via noncovalent interactions. The asymmetric catalytic investigation by this nanotubular structure is underway in our laboratory.

## Methods

All starting reagents and solvents were used as commercially purchased from J&K scientific without further purification unless otherwise noted (AR, 99.7%). Nuclear magnetic resonance (NMR) spectra ($^1$H NMR, COSY, DOSY, NOESY) were measured on a Bruker AVANCE III 400 (400 MHz) spectrometer with $^1$H chemical shifts quoted in ppm relative to the signals of the residual non-deuterated solvents or 0.0 ppm for tetramethyl silane (TMS). HRESI-TOF mass spectra were recorded on Bruker maXis 4 G. The data analyses of ESI-TOF MS were processed on Bruker Data Analysis software and the simulations were performed on Bruker Isotope Pattern software. Single crystal X-ray diffraction data were collected on an Agilent SuperNova X-ray diffractometer using micro-focus X-ray sources (Cu-$K_\alpha$, $\lambda = 1.54184$ Å). Powder X-ray diffraction (PXRD) was carried out with a Rigaku SmartLab diffractometer (Bragg-Brentano geometry, Cu-$K_{\alpha 1}$ radiation, 40 kV and 30 mA, $\lambda = 1.54056$ Å). AFM measurements were carried out on Dimension FastScan Bio atomic force microscope using a mica plate as substrate. Transmission electron microscopy (TEM) measurements were carried out on FEI Tecnai G2 F30 (300 kV). Isothermal Titration Calorimetry (ITC) experiments were carried out in aqueous solution (DMSO/H$_2$O = 1/2, v/v) at 25 °C on a VP-ITC instrument (Malvern MicroCal VP-ITC). Enantioselectivity analysis was measured on an Agilent 7820 A GC system equipped with a Supelco β-DEX 120 silica capillary column (30 m × 0.25 mm × 0.25 μm film thickness).

## Data availability

The data generated in this study are provided in this article and Supplementary Information file. Additional data are available from the corresponding author upon request. Crystallographic data for R/S-nanotube have been deposited at the Cambridge Crystallographic Data Centre (www.ccdc.cam.ac.uk/data_request/cif) and could be downloaded free of charge by the deposition number of 2258255 and 2258256, respectively.

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

## Acknowledgements

This work was supported by the Guangdong Province Key Field R&D Plan Project (2020B0101030005 to Y.P.C.), the Guangdong Basic and Applied Basic Research (2022A1515012057 to K.L.), Guangzhou Basic and Applied Basic Research (202201011342 to K.L.).

## Author contributions

K.L. and Y.P.C. conceived and directed the project. W.M.Q., W.X.S. and J.M.H. carried out the synthesis and characterizations. K.L. performed the single-crystal X-ray analysis and catalytic investigation. K.L. and Y.P.C. co-wrote the manuscript and all authors discussed the results and commented on the paper.

## Competing interests

The authors declare no competing interests.
