## [Peer Review File · Nature Communications]

REVIEWER COMMENTS

Reviewer #1 (Remarks to the Author):

I read with interest this article by Cai, Li and co-workers on the catalysis the 3-aza-Cope rearrangement by cyclic supramolecular oligomers of a binaphthyl based phosphoric ester. Molecules of such esters assemble in DMSO/Water when NaOH is added to give supramolecular structures, which are likely formed by the spatial superimposition of 6-membered supramolecular cyclo-oligomers. X-Ray crystal analysis show that in the solid state, that such structure are in fact seen, although much more long and organized into long pseudo tubes. Many experimental pieces of evidence have been collected to support the hypothesized association model in solution. It is a great pity that the nanotubes does not work in heterogenous conditions. This would have given to the manuscript a higher significance. I think that the manuscript has the potential to be published in a highly ranked journal like Nature Communications but a major revision indeed in order to address the issues listed below.

1) I have to say that it is very hard to digest that deprotonation by NaOH, which generates negative charges, increases the hydrophobicity of the cat 1, causing the assembly cat 1 monomers. The author should explain better this point, which is not intuitive.

2) Concerning the titration with ITC. What is the binding model to which the authors refer to? An association constant given in M-1 calls for a 1:1 model. But in the titration experiment the system is in assembly conditions. Not only but the author claim that the short nanotube hosts three molecules of substrate... It is certainly not described by a 1:1 model providing an association constant measured in M-1. And also, do the substrate react during the titration?

3) Figure 3c right: are the distance reported in x-axis right? (200-300 nm compare badly with 3 nm). Furthermore it is not so clear for me how the AFM experiment can contribute to reinforce the structure on the short nanotubes. The hypothesis given by the authors is that such nanotubes are placed side by side with each other so that the length of the fibers observed is the height of the of the short nanotube (3 nm). How can the authors be sure that the short nanotubes are arranged side by side looking at the image in Figure 3c, left?

Figure 3e; what is the meaning of the curve?. Is it cosmetic or calculated on the basis of a model-equation?

Minor points:

1) caption to Figure 3c, substitute aqueous solution with the more precise DMSO/water 1:2

2) Table 1, substitute 16.7 with 17 and 85.2 with 85

3) I would suggest to remove the section concerning the Arrhenius activation energy, which is little informative.

4) regarding the section on Eyring activation parameters, I like the argument given by the authors. I would suggest to cite Eur. J. Org. Chem. 2014, 7304–7315, which surely strengthens the interpretation of the experimental results given by the authors.

Reviewer #2 (Remarks to the Author):

The paper describes a study focused on creating supramolecular confined catalysis by constructing a hexagonal nanotubular structure using chiral 1,1'-bi-2-naphthol (BINOL) phosphates in an aqueous medium. The nanotube's structure is characterized through various techniques, and its stability in a homogeneous aqueous solution is confirmed. The length of the nanotube in the solution is found to depend on the concentration of chiral BINOL-phosphate. The study demonstrates that the assembled nanotube accelerates the rate of the 3-aza-Cope rearrangement reaction significantly, up to 85-fold. This acceleration is attributed to the interior confinement effect, where chain-like substrates are constrained and pre-organized into a reactive chair-like conformation within the nanotube. The proposed mechanism suggests that this pre-organization stabilizes the transition state of the reaction, leading to a more pronounced acceleration effect.

Supramolecular confined catalysis through the construction of a nanotubular structure is an interesting research field and the applied various characterization techniques, including X-ray, NMR, ESI-MS, AFM, and TEM, are sufficient. The demonstrated acceleration of the reaction suggests potential practical applications in catalysis. However, accelerated reactions in various confined spaces are very well known. The challenge is still to find examples with high asymmetric induction. The paper can be published in Nature Communications after the following has been addressed. The author state that they use a pair of enantiomers. They should write for this "pair" the correct term "racemate". Furthermore, the authors state that they currently are working on an asymmetric version, however they should already include an example of an asymmetric version in this paper. What happens if they use enantiopure BINAP. Racemates and pure enantiomers can behave very differently. Are nanotubes still formed with enantiopure compounds and if yes, they should provide one example asymmetric version of the aza-Cope rearrangement. Would a significant ee be observed?

Point by Point Response to the Reviewer's Comments

Reviewers' comments:

Reviewer #1 (Remarks to the Author):

Comment: *"I read with interest this article by Cai, Li and co-workers on the catalysis the 3-aza-Cope rearrangement by cyclic supramolecular oligomers of a binaphthyl based phosphoric ester. Molecules of such esters assemble in DMSO/Water when NaOH is added to give supramolecular structures, which are likely formed by the spatial superimposition of 6-membered supramolecular cyclo-oligomers. X-Ray crystal analysis show that in the solid state, that such structure are in fact seen, although much more long and organized into long pseudo tubes. Many experimental pieces of evidence have been collected to support the hypothesized association model in solution. It is a great pity that the nanotubes does not work in heterogenous conditions. This would have given to the manuscript a higher significance. I think that the manuscript has the potential to be published in a highly ranked journal like Nature Communications but a major revision indeed in order to address the issues listed below."*

Response: Thank you for your recognition of our work and supportive feedback. We have addressed your concerns below.

Comment: *"1) I have to say that it is very hard to digest that deprotonation by NaOH, which generates negative charges, increases the hydrophobicity of the cat 1, causing the assembly cat 1 monomers. The author should explain better this point, which is not intuitive."*

Response: We agree with the hydrophobicity issue you have raised for the assembly process. The deprotonated cat1 monomer should be more hydrophilic rather than hydrophobic in our case. It's improper to adopt the hydrophobic effect to explain the self-assembly process. In the revised version, we deleted the hydrophobic discussion and mainly attributed it to the geometry-directed intermolecular recognition and arrangement in aqueous medium as demonstrated in Figure 1b and highlighted the revised text in yellow.

Comment: “2) Concerning the titration with ITC. What is the binding model to which the authors refer to? An association constant given in M-1 calls for a 1:1 model. But in the titration experiment the system is in assembly conditions. Not only but the author claim that the short nanotube hosts three molecules of substrate... It is certainly not described by a 1:1 model providing an association constant measured in M-1. And also, do the substrate react during the titration?”

Response: According to the ITC titration analysis, it appeared to be a 1:3 binding model for both substrates and a multi-site fitting should be applied to give the three successive binding constants. However, there is only one titration plateau existed in the ITC profiles for both substrates, which means the stepwise binding of three guests is completely overlapped. This condition is difficult for a multi-site fitting and would give serious error, especially beyond the 1:2 model. Actually, we have tried the multi-site binding model but didn't get the satisfied result. To simplify the mathematical process, we used the 1:1 model to fit the binding constant (K_a) that is considered as the general parameter to represent the whole binding process. This treatment is similar to the Hill-equation method (*Arch. Hist. Exact Sci.* **2012**, 427–438), which is widely used for protein and substrate binding interactions. In this method, no matter how many substrates as ligand binding to the protein site, the equation gives only one K_a parameter (M^{-1}) as an average value to describe the overall host-guest interactions.

For the issue of whether the substrate reacting during the titration raised, the answer is yes. According to the kinetic analysis result of reaction rate under the nanotube conditions, the reaction rate constants for S1 and S2 at room temperature (25 °C) are $2.5 \times 10^{-5} s^{-1}$ and $3.0 \times 10^{-4} s^{-1}$, respectively. Based on these two rate values, we could calculate out that about 8 % for S1 and 65 % for S2 converted to their corresponding products after 1 h, which time the ITC titration takes in our original experiment. For S1, this little conversion would have less effect on the titration process and analysis. However, the negative influence of the most conversion for S2 cannot be ignored. To address this problem and obtain a more accurate binding constant for S2, we reduced the amount of data point collection and shortened the testing time to suppress the substrate conversion during ITC experiment. In practice, the whole titration time was controlled within 15 min and a plateau appeared in the titration profile in about 6 min that means the nanotube as host is saturated by the substrate binding. During this period of time (6

min), the conversion of S2 was reduced to about 10 % and a more reliable binding constant was obtained ($9.41 \times 10^4 \text{ M}^{-1}$ versus the former $5.94 \times 10^4 \text{ M}^{-1}$). The new binding constant and ITC titration profile for S2 has been updated and highlighted in yellow in revised article and SI.

Comment: “3) Figure 3c right: are the distance reported in x-axis right? (200-300 nm compare badly with 3 nm). Furthermore it is not so clear for me how the AFM experiment can contribute to reinforce the structure on the short nanotubes. The hypothesis given by the authors is that such nanotubes are placed side by side with each other so that the length of the fibers observed is the height of the of the short nanotube (3 nm). How can the authors be sure that the short nanotubes are arranged side by side looking at the image in Figure 3c, left? Figure 3e; what is the meaning of the curve?. Is it cosmetic or calculated on the basis of a model-equation?”

Response: Thank you for your carefulness to check our AFM results. As you mentioned about the horizontal size (x-axis, ca. 100 nm) matched badly with the vertical size (y-axis, 3 nm) for the assembled nanotube in cutting profile from the AFM image, actually, the vertical size is more adopted as the height of sample rather than the horizontal size as width in AFM analysis. Because the tip of probe needle that contact with the sample surface has its own width as suggested by the technician, it would disturb the measurement along the horizontal direction and cause significant error. Therefore, compared with the horizontal size, the vertical size of height is a more reliable value that can represent the real size of material. Meanwhile, we have found two references (*Nat. Chem.*, **2022**, 507-514; *J. Am. Chem. Soc.*, **2024**, 450-459) that both used the AFM technology to characterize their nanotube structures. In *ref. 1* (Figure 1a), the covalent-organic framework (COF) based single nanotube has a radial size of 4.3 nm that matches well with the vertical height of AFM image (ca. 5 nm), while the horizontal size even reaches to micrometer (μm) scale. Likewise in *ref. 2* (Figure 1b), the vertical size of height (ca. 7 nm) is in accordance with the radial size of polyoxometalate (POM) based single nanotube (7.1 nm), in opposite, the horizontal size reaches to 100 nm scale that largely exceeds the real width of the nanotube structure.

Figure 1. AFM images of COF-based (a) and POM-based (b) single nanotubes as reported by literatures.

Besides the AFM result that gave the height of the single nanotube to be 3 nm, we have obtained and solved the single crystal structure of the packed nanotubes in periodic lattice. The radial size of the single nanotube was measured to be 3.0 nm from the X-ray crystal structure analysis (see Figure 2c), which was entire in accordance with the AFM result. This part of discussion could be found in page 5, line 24 and 25. This coincidence can be attributed to the in situ scanning the sample through AFM probe needle dipping into the nanotube solution, which maintains the integrity of the single nanotube morphology under wet conditions and avoids the stacking and aggregation once drying out from the solution. Therefore, it can be concluded that the nanotubes in AFM image are arranged side by side as you have mentioned.

For Figure 3e you have mentioned, it shows the gradual growth of the single nanotube's length in solution with the increase of NOP⁻ monomer's concentration. The curve was plotted based on five data points that were obtained from the ¹H DOSY measurement and calculations using two model-equations. In this graph, to better interpret each data point, we have drawn the stacked torus with different layers (marked number) to represent the growth of single nanotube. The first two stacked torus of two layers were drawn as it to be. The last three were not drawn according to their real layer number because of the relatively large values (7-13-21) and simplified for clarity purpose. Thus, we

have supplemented additional description in the caption of Figure 3e and highlighted in yellow.

Comment: *Minor points:*

1) *caption to Figure 3c, substitute aqueous solution with the more precise DMSO/water 1:2*

Response: We have substituted it as you suggested and highlighted in yellow.

2) *Table 1, substitute 16.7 with 17 and 85.2 with 85*

Response: We have substituted them both in Table 1 as you suggested and highlighted in yellow.

3) *I would suggest to remove the section concerning the Arrhenius activation energy, which is little informative.*

Response: We have removed all the results and discussions about Arrhenius activation energy in article and supplementary information files as you suggested.

4) *regarding the section on Eyring activation parameters, I like the argument given by the authors. I would suggest to cite Eur. J. Org. Chem. 2014, 7304–7315, which surely strengthens the interpretation of the experimental results given by the authors.*

Response: We have cited this paper as *ref. 50* in page 9, line 23 and updated the reference list.

Reviewer #2 (Remarks to the Author):

Comment: *“The paper describes a study focused on creating supramolecular confined catalysis by constructing a hexagonal nanotubular structure using chiral 1,1'-bi-2-naphthol (BINOL) phosphates in an aqueous medium. The nanotube's structure is characterized through various techniques, and its stability in a homogeneous aqueous solution is confirmed. The length of the nanotube in the solution is found to depend on the concentration of chiral BINOL-phosphate. The study demonstrates that the assembled nanotube accelerates the rate of the 3-aza-Cope rearrangement reaction significantly, up to 85-fold. This acceleration is attributed to the interior confinement effect, where chain-like substrates are constrained and pre-organized into a reactive chair-like conformation within the nanotube. The proposed mechanism suggests that this pre-organization stabilizes the transition state of the reaction, leading to a more*

pronounced acceleration effect.”

Response: Thank you for your reading and summary of our work. We have addressed your concerns below.

Comment: *“Supramolecular confined catalysis through the construction of a nanotubular structure is an interesting research field and the applied various characterization techniques, including X-ray, NMR, ESI-MS, AFM, and TEM, are sufficient. The demonstrated acceleration of the reaction suggests potential practical applications in catalysis. However, accelerated reactions in various confined spaces are very well known. The challenge is still to find examples with high asymmetric induction. The paper can be published in Nature Communications after the following has been addressed. The author state that they use a pair of enantiomers. They should write for this “pair” the correct term “racemate”. Furthermore, the authors state that they currently are working on an asymmetric version, however they should already include an example of an asymmetric version in this paper. What happens if they use enantiopure BINAP. Racemates and pure enantiomers can behave very differently. Are nanotubes still formed with enantiopure compounds and if yes, they should provide one example asymmetric version of the aza-Cope rearrangement. Would a significant ee be observed?”*

Response: You mentioned about the expression of “a pair of enantiomers” in this article. Actually, we used the chiral form of BINOL-phosphate rather than the racemate to assemble the nanotubular structure and obtained two corresponding chiral nanotubes (denoted as R-nanotube and S-nanotube) that were unambiguously confirmed by single crystal X-ray analysis (crystallized in chiral space group). To avoid any misunderstanding caused by this expression, we deleted the phrase of “a pair of” associated with enantiomers in this article.

As you have mentioned, the chiral nanotubes as catalysts should have function for asymmetric catalysis. Following your suggestion, we synthesized another substrate (S3) that can generate a chiral carbon center after catalytic transformation. The catalytic results gave a moderate enantioselectivity (around 43% ee value) for S3 under R-nanotube or S-nanotube existing conditions, which showed great potential and improved space for the next asymmetric catalytic investigation. We have added the related discussion in page 11 and supplemented data

in SI with yellowed highlight.

REVIEWERS' COMMENTS

Reviewer #1 (Remarks to the Author):

The authors addressed all my concerns and I am pleased to support publication of the manuscript in Nature Communications.

Reviewer #2 (Remarks to the Author):

The revised paper has addressed all comments of the reviewers and the additional work has improved the manuscript significantly. Hence, the work can be published in nature communications.